# Mixture-of-Experts for Knowledge Graph Retrieval-Augmented Generation

## Abstract

Large Language Models (LLMs) have demonstrated strong capabilities in open-domain question answering, but often struggle with factual accuracy and multi-hop reasoning due to the incompleteness of the training corpora. A promising solution is Knowledge Graph Retrieval-Augmented Generation (KG-RAG), which supplements LLMs with structured knowledge retrieved from external knowledge graphs (KGs). However, existing KG-RAG methods either rely on large-scale language models (e.g., ChatGPT) to guide the retrieval process, which leads to high computational costs, or suffer from limited retrieval quality when using lightweight language models, particularly under multi-hop scenarios. We propose MoRA (Mixture-of-Experts for Retrieval-Augmented Generation over Knowledge Graphs), a novel KG-RAG framework that enhances hop-wise KG knowledge retrieval through a Mixture-of-Experts (MoE) framework. Each expert is guided by a combination of two types of soft prompts: *expert-specific soft prompt* encourages specialization in different reasoning perspectives across experts, and *contextual soft prompt* evolves with each reasoning hop by encoding the query and previously explored KG triplets, enabling the model to preserve consistency and relevance across multi-hop retrieval. This design allows MoRA to perform accurate and robust retrieval using lightweight language models. MoRA achieves superior performance across multiple KG-based Question Answering benchmarks compared to existing retrieval systems, including those that rely on much larger language models, demonstrating its effectiveness under limited computational budgets.

## 1 Introduction

Large Language Models (LLMs) (Brown et al., 2020; Touvron et al., 2023a) have demonstrated impressive capabilities across a wide range of natural language processing tasks, including text comprehension (Lewis et al., 2020a), open-domain question answering (Wei et al., 2022b), and natural language generation (Cheng et al., 2023). Despite these advances, LLMs remain limited due to the incompleteness of knowledge encoded during pretraining, which restricts their ability to provide factually accurate and up-to-date answers (Zheng et al., 2023; Wang et al., 2024). This limitation is particularly pronounced in knowledge-intensive scenarios that demand factual precision, reasoning over complex multi-hop relations, and adaptation to continually evolving information (Zheng et al., 2023; Wang et al., 2024). Consequently, LLMs often generate responses that contain hallucinated content, outdated facts, or unverifiable claims, undermining their reliability in real-world applications.

To mitigate these issues, Retrieval-Augmented Generation (RAG) (Gao et al., 2023b) has been proposed to enhance LLM reliability by incorporating external knowledge during inference. RAG allows models to access up-to-date and factual information beyond what was memorized during pretraining. While early RAG systems primarily relied on unstructured text corpora, recent efforts have explored structured sources of knowledge for more precise and interpretable reasoning. Among these, Knowledge Graphs (KGs) offer a highly structured and semantically rich format, representing facts as *triplets* in the format of $(\text{subject}, \text{relation}, \text{object})$. This format supports explicit relation modeling and multi-hop relational reasoning, making KGs especially suitable for question answering tasks that require factual consistency and transparency (Sun et al., 2024; Jiang et al., 2025). In the KG-RAG paradigm (Sanmartin, 2024), models typically follow a two-step process: (1) *knowledge retrieval*, where a subgraph containing query-relevant triplets is constructed by expanding from entities appearing in the query; and (2) *prompt integration*, where triplets in the retrieved subgraph are presented as input to the LLM as evidence for answer generation. The performance of KG-RAG

methods largely relies on the ability to retrieve and rank relevant triplets effectively, especially under multi-hop settings where irrelevant information can accumulate and mislead reasoning.

Despite recent progress, current KG-RAG approaches still face limitations that hinder their effectiveness. First, most methods assess candidate triplet relevance through a single lens, such as semantic similarity or prompt-based scoring (Shi et al., 2024; Sun et al., 2024). This rigid design fails to accommodate the diverse reasoning demands across different semantic types of triplets, lacking the flexibility to account for diverse relational semantics (Saxena et al., 2020). A unified reasoning strategy cannot effectively handle the diversity, leading to suboptimal triplet selection. Second, many candidate triplets may appear irrelevant when judged in isolation, especially those involving underspecified relations, but actually play a critical role in connecting earlier reasoning steps to the final answer. Existing methods typically treat each KG triplet independently and fail to model such multi-hop dependencies (Feng et al., 2020; Zhang et al., 2022). As a result, they struggle to maintain contextual continuity, often discarding useful evidence prematurely. Third, while most recent systems attempt to address these issues by leveraging large-scale language models such as ChatGPT to guide retrieval, which are computationally expensive and impractical for scalable or real-time applications (Min et al., 2019). Examples of the first two limitations are shown in Figure 1.

To address these challenges, we propose MoRA, a lightweight but effective KG-RAG framework that enables lightweight language models to perform accurate and multi-hop retrieval. MoRA contains a *Mixture-of-Experts* (MoE) scoring module, which scores candidate KG triplets based on the query and previously explored knowledge. Each expert in the MoE is controlled by a learnable *expert-specific soft prompt*, encouraging specialization in reasoning styles such as temporal reasoning or compositional logic. In addition, MoRA incorporates a *contextual soft prompt* that is dynamically constructed from the triplets in previous reasoning steps. This prompt provides evolving contextual signals that help the scoring of new candidate triplets using earlier reasoning steps, allowing the model to assess each candidate not in isolation, but in light of the evidence already accumulated, addressing the limitation of prior work. By combining the semantic diversity of multiple experts with contextual adaptation, our model robustly filters irrelevant or noisy triplets. Notably, this framework is fully compatible with lightweight models, avoiding the reliance on expensive LLMs such as ChatGPT for retrieval, thereby offering both efficiency and strong performance. Our main contributions are listed as follows:

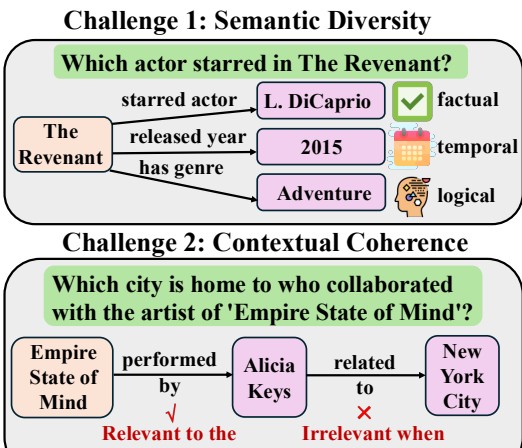

Figure 1: Illustration of two key challenges in KG-RAG. In Challenge 1, candidate triplets from the same entity span different semantic directions: factual, temporal, or logical, requiring expert specialization for effective scoring. In Challenge 2, a relation like `related to` may appear irrelevant when judged in isolation, but becomes crucial when combined with prior steps.

- We propose MoRA, a KG-RAG framework based on Mixture-of-Experts that enables lightweight language models to conduct accurate multi-hop KG retrieval, reducing the cost of retrieval.
- We design two types of soft prompts to support our reasoning: (1) *expert-specific soft prompt*, which allows different experts to focus on distinct aspects of reasoning, and (2) *contextual soft prompt* that incorporates evidence from the prior hop to support contextual retrieval. This combination brings both specialization and adaptivity into the retrieval process.
- We conduct extensive experiments on four KG-based QA benchmark datasets. MoRA consistently outperforms existing KG-RAG baselines across all settings. Remarkably, it even surpasses *ToG (ChatGPT-3.5)* despite using only lightweight language models for retrieval, achieving better accuracy and lower inference cost.

## 2 PROBLEM FORMULATION

We study the task of Knowledge Graph Retrieval-Augmented Generation (KG-RAG), where structured knowledge from a knowledge graph is selectively retrieved to assist a language model in

answering natural language questions. Formally, a knowledge graph is represented as $\mathcal{G} = (\mathcal{E}, \mathcal{R}, \mathcal{T})$, where $\mathcal{E}$ denotes the set of entities, $\mathcal{R}$ is the set of relations, and $\mathcal{T} = \{(h, r, t) \mid h, t \in \mathcal{E}, r \in \mathcal{R}\}$ is the set of directed triplets. Each triplet $(h, r, t)$ encodes the fact that head entity $h$ is connected to tail entity $t$ via relation $r$. Given a language model and a question $q$, the objective is to identify a subset of triplets from $\mathcal{T}$ that are relevant to answering $q$, and use them as evidence to enhance the generation of LLM. Formally, if $\mathcal{T}_q^* \subseteq \mathcal{T}$ denotes the selected triplet set, the final answer $a_q$ is generated as:

$$a_q = \text{LLM}\big([q; \mathcal{T}_q^*]\big). \tag{1}$$

Here $\mathcal{T}_q^*$ contains the most relevant information for question answering.

## 3 MoRA

We propose MoRA, a compact and adaptive framework for Knowledge Graph Retrieval-Augmented Generation (KG-RAG), designed to enhance multi-hop reasoning and knowledge retrieval using lightweight language models. MoRA operates in a multi-hop retrieval setting, where relevant triplets from the knowledge graph are selected iteratively based on both the input query $q$ and the reasoning context of explored knowledge. As shown in Figure 2, the framework introduces two key components: (1) a *Mixture-of-Experts (MoE)* scoring module, where multiple experts guided by distinct *expert-specific soft prompts* specialize in different semantic perspectives; and (2) *contextual soft prompts* that are dynamically constructed from the query and previously explored knowledge, enabling hop-aware retrieval and maintaining coherence across multi-hop reasoning steps. These components work together to provide a diverse and context-sensitive evaluation of candidate triplets, allowing MoRA to robustly retrieve useful knowledge even without relying on LLMs with enormous scales.

### 3.1 Multi-Hop Retrieval Framework

Given a knowledge graph $\mathcal{G}$ and a natural language question $q$, our objective is to retrieve a set of triplets $\mathcal{T}_q^*$ from $\mathcal{G}$ that provide faithful evidence for answering $q$. To achieve this, we design an iterative retrieval framework that incrementally explores the KG in a step-wise manner.

**Step 1: Candidate Construction.** To initiate retrieval, we first identify a set of query entities $\mathcal{E}_q$ mentioned in the question $q$, and initialize the first-hop frontier as $\mathcal{E}^{(0)} = \mathcal{E}_q$. At the $t$-th retrieval hop, we construct a set of candidate triplets $\mathcal{C}^{(t)}$ by expanding outward from the current frontier $\mathcal{E}^{(t-1)}$. Specifically, we include all outgoing triplets $(e, r, e')$ for $e \in \mathcal{E}^{(t-1)}$, as well as reversed triplets $(e, r^{-1}, e')$ whenever $(e', r, e)$ appears in the KG:

$$\mathcal{C}^{(t)} = \{(e, r, e') \mid e \in \mathcal{E}^{(t-1)}\} \cup \{(e, r^{-1}, e') \mid (e', r, e) \in \mathcal{G}, \ e \in \mathcal{E}^{(t-1)}\}. \tag{2}$$

This candidate construction step ensures broad coverage of possible evidence paths, supporting both forward and backward reasoning chains. The resulting set $\mathcal{C}^{(t)}$ serves as the input to the scoring module of hop $t$ described in the next step.

**Step 2: Mixture-of-Experts Scoring and Triplet Selection.** Each candidate triplet $x \in \mathcal{C}^{(t)}$ is evaluated for its relevance to the question $q$ and the context knowledge explored before. To model different reasoning patterns and maintain contextual consistency across hops, we employ a modular scoring mechanism based on a Mixture-of-Experts architecture with soft prompts (detailed in subsection 3.2). This module outputs a contextualized score $S(q, x, t)$ for each candidate. We retain the top-$M$ triplets by $S(q, x, t)$ for further exploration:

$$\mathcal{T}_{\text{sel}}^{(t)} = \text{Top-}M\{S(q, x, t) \mid x \in \mathcal{C}^{(t)}\}. \tag{3}$$

This step prioritizes the relevant triplets while controlling the expansion size and noise accumulation. We maintain a visited set $\mathcal{V}^{(t)}$ to track explored entities. At the beginning, we initialize $\mathcal{T}_q^* = \emptyset$ and $\mathcal{V}^{(0)} = \mathcal{E}^{(0)}$. At each hop $t$, the selected triplets $\mathcal{T}_{\text{sel}}^{(t)}$ are added to the evidence set, and the set of new tail entities is set as the frontier for the next hop:

$$\mathcal{T}_q^* \leftarrow \mathcal{T}_q^* \cup \mathcal{T}_{\text{sel}}^{(t)}, \tag{4}$$

$$\mathcal{E}^{(t)} = \{e' \mid (h, r, e') \in \mathcal{T}_{\text{sel}}^{(t)}\} \setminus \mathcal{V}^{(t-1)}, \tag{5}$$

$$\mathcal{V}^{(t)} = \mathcal{V}^{(t-1)} \cup \mathcal{E}^{(t)}. \tag{6}$$

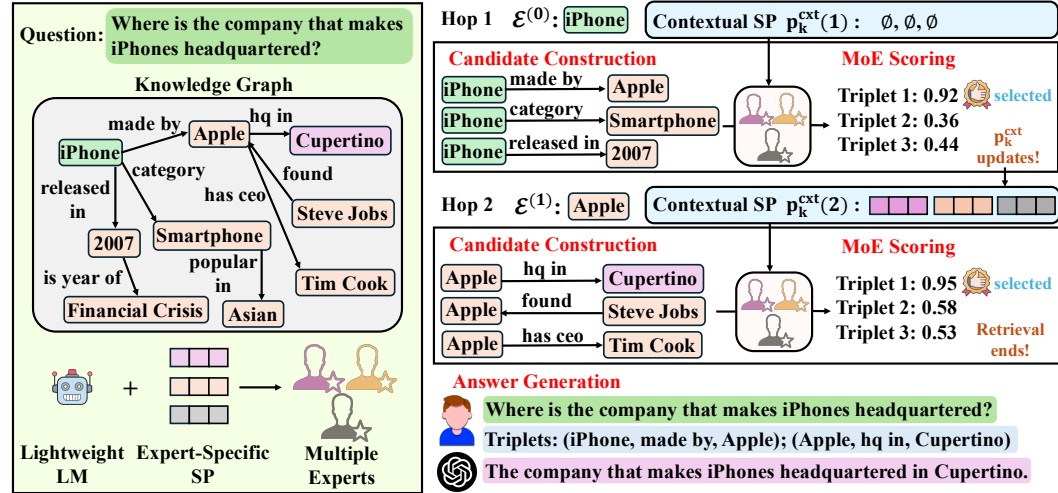

Figure 2: Overview of MoRA framework for KG-RAG. The framework consists of: (1) Candidate Construction: From the frontier entities $\mathcal{E}^{(t-1)}$, initially the entities in question $q$, the framework iteratively expands the candidate triplet set $\mathcal{C}^{(t)}$ from the KG. (2) Mixture-of-Experts Scoring: Candidate triplets are evaluated by a lightweight language model (LM) with two types of soft prompts (SP): Expert-specific soft prompt specializes in different experts on distinct semantic perspectives. Contextual soft prompt is updated across hops to propagate reasoning history and maintain coherence during multi-hop inference. Expert predictions are combined into a final score for each candidate. Top-scoring triplets are selected at each hop and accumulated into the evidence set. (3) Answer Generation: The collected evidence is provided to a large generator for answer generation.

By keeping track of visited nodes, we avoid redundancy and loops.

**Iterative Retrieval.** The above process is repeated for a fixed number of hops $H$, progressively building an evidence graph rooted in the query. The final triplet set $\mathcal{T}_q^*$ contains the most relevant multi-hop facts and is passed to the downstream answer generator.

Unlike methods that rely on large-scale language models to implicitly model reasoning chains, our framework performs retrieval in an explicit and structured way using only lightweight language models for triplet scoring. This enables efficient and scalable multi-hop retrieval with better interpretability and lower cost. The design of our scoring module is key to enabling such structured reasoning and will be detailed in the next subsection.

## 3.2 Mixture-of-Experts Scoring

To robustly evaluate candidate triplets in multi-hop retrieval, we design a compact neural scoring module based on a Mixture-of-Experts (MoE) architecture with contextualized soft prompts. A *soft prompt* refers to a learnable continuous embedding prepended to the input of a language model, allowing efficient task adaptation without fine-tuning the model parameters. This module addresses two key challenges in KG-RAG by applying different soft prompts: (1) capturing diverse semantic reasoning patterns, and (2) maintaining contextual coherence across hops. To this end, each expert is equipped with two types of trainable soft prompts: an *expert-specific soft prompt* that encodes the expert's semantic bias, and a *contextual soft prompt* that evolves over retrieval steps by summarizing the expert's prior output representations. These components allow the model to specialize across different reasoning types while remaining sensitive to context evolution across hops.

Each expert operates as an independent scorer built on a shared, lightweight language model. Given a question $q$ and a candidate triplet $x$ at hop $t$, the $k$-th of $K$ experts constructs its input by concatenating the *expert-specific soft prompt* $p_k^{\text{exp}}$, the *contextual soft prompt* $p_k^{\text{cxt}}(t)$, the question $q$, and a natural language serialization of the candidate triplet $x$ to obtain a new output embedding:

$$\mathbf{z}_k(q, x, t) = \text{LM}\big([p_k^{\text{cxt}}(t); \ p_k^{\text{exp}}; \ q; \ x]\big), \tag{7}$$

where LM denotes the lightweight language model shared across all experts. $p_k^{\text{exp}}$ is a learnable embedding sequence that is unique to expert $k$ and $p_k^{\text{cxt}}(t)$ encodes knowledge from last hop. $z_k(q, x, t)$ encodes relevance under the expert's specialization and the current reasoning state.

To maintain temporal continuity, the *contextual soft prompt* is updated after each hop by aggregating expert outputs from the previous step:

$$p_k^{\text{cxt}}(t) = g\left(\frac{1}{|\mathcal{C}^{(t-1)}|} \sum_{x \in \mathcal{C}^{(t-1)}} \mathbf{z}_k(q, x, t-1)\right),\tag{8}$$

where $g(\cdot)$ is a learned projection MLP layer. For the first hop ($t = 1$), this prompt is omitted in the input of LM. This update allows each expert to track their own reasoning path across multiple retrieval steps. The $k$-th expert transforms its embedding into a scalar score $s_k(q, x, t) \in (0, 1)$ with a learnable linear layer $\mathbf{w}_k$ and a sigmoid layer $\sigma$:

$$s_k(q, x, t) = \sigma(\mathbf{w}_k^\top \mathbf{z}_k(q, x, t)),\tag{9}$$

and a soft routing network assigns mixture weights to each expert. Specifically, the lightweight language model encoder produces a representation of the question and candidate as $f(q, x)$, and a trainable router matrix $W_r$ maps this to a weight distribution on $K$ experts for triplet $x$:

$$\alpha(q, x, t) = \text{softmax}(W_r \cdot f(q, x)).\tag{10}$$

The final score for candidate $x$ is the weighted sum over experts:

$$S(q, x, t) = \sum_{k=1}^{K} \alpha(q, x, t)[k] \cdot s_k(q, x, t).\tag{11}$$

This scoring architecture directly addresses the key challenges in multi-hop KG retrieval. The *expert-specific soft prompts* enable each expert to specialize in different semantic patterns, such as type constraints or temporal dependencies, ensuring flexible matching of different query intents and knowledge types in tasks. The *contextual soft prompts* preserve reasoning continuity across hops by incorporating step-wise retrieval history into scoring. Together, these components allow the retriever to adapt its behavior based on both the question semantics and the evolving context. Importantly, the entire module operates over lightweight language models with soft prompt tuning, offering an efficient and scalable alternative to LLM-based retrievers.

### 3.3 ANSWER GENERATION

After the multi-hop selection process, we obtain the final evidence triplet set $\mathcal{T}_q^*$ that aggregates promising triplets across hops. To generate the final answer, we leverage a powerful large-scale language model, such as ChatGPT, which excels at reasoning over textualized knowledge. Each triplet $x$ is first serialized into a natural language sentence. The evidence set is then concatenated into a textual context. The LLM receives the question $q$ and the retrieved evidence $\mathcal{T}_q^*$ as input:

$$a_q = \text{LLM}\big([q; \mathcal{T}_q^*]\big),\tag{12}$$

where $a_q$ denotes the generated answer. Since the LLM is conditioned on the selected evidence, it grounds its reasoning on relevant multi-hop chains rather than the entire KG.

While this generation step can leverage powerful LLMs, our framework is designed to minimize reliance on them. Due to the MoRA retriever's MoE scoring mechanism built from compact language models and soft prompts, the system can already filter highly relevant and coherent evidence chains. This allows downstream generation to focus on language fluency, rather than compensating for missing knowledge or incorrect reasoning paths.

By explicitly addressing key challenges in KG-RAG, such as semantic diversity (via expert-specific soft prompts), cross-hop consistency (via contextual soft prompts), and LLM scale dependency (via lightweight model retrieval), our approach provides a scalable and interpretable solution. The triplet evidence offers transparent reasoning traces, while the generation stage remains modular and adaptable to different deployment needs.

### 3.4 TRAINING OBJECTIVE

Our learnable components in MoRA are trained by a contrastive learning process. Importantly, the underlying pretrained language models (both the lightweight language model and the large answer

generator) remain frozen throughout training. This ensures that learning focuses on adapting the lightweight modules for effective scoring and information summarization.

Supervision is provided by question–answer pairs together with ground-truth subgraphs. For a given question $q$, in the hop $t$, let $\mathcal{T}_{q,t}^{+} \in \mathcal{C}^{(t)}$ denote the set of gold triplets that lie on reasoning paths leading a query entity in $\mathcal{E}_q$ to any answer entity within a predefined path length, and let $\mathcal{T}_{q,t}^{-} \in \mathcal{C}^{(t)}$ denote triplets sampled from the candidates that are not part of any gold path. The objective is to assign higher scores to $\mathcal{T}_{q,t}^{+}$ while suppressing $\mathcal{T}_{q,t}^{-}$.

We adopt a cross-entropy loss over candidate triplets at hop $t$:

$$\mathcal{L} = -\frac{1}{|\mathcal{T}_{q,t}^{+}|} \sum_{x \in \mathcal{T}_{q,t}^{+}} \log(S(q,x,t)) - \frac{1}{|\mathcal{T}_{q,t}^{-}|} \sum_{x \in \mathcal{T}_{q,t}^{-}} \log(1 - S(q,x,t)), \tag{13}$$

All trainable parameters are optimized jointly by backpropagation. Through this objective, the scorer learns to emphasize relevant triplets and suppress distractors, providing reliable multi-hop evidence for the downstream answer generator.

## 4 EXPERIMENTS

In this section, we evaluate the effectiveness of our proposed approach across multiple QA benchmark datasets. We compare our method against several baselines to assess its ability to retrieve and utilize knowledge during KG-RAG for improving LLM-based question answering.

### 4.1 DATASETS

We evaluate our method on four Question Answering datasets: WebQuestionsSP (WebQSP) (Yih et al., 2016), WikiMovies (Miller et al., 2016), and the datasets of MetaQA (Zhang et al., 2018) in both 1-hop and 2-hop settings. WebQSP contains 4,737 natural language questions annotated with corresponding SPARQL queries over Freebase (Bollacker et al., 2008), a broad-coverage knowledge graph. This dataset requires both single- and multi-hop structured reasoning and is commonly used to evaluate complex KG-based QA. WikiMovies is a domain-specific dataset focused on the movie domain, constructed from WikiData and Wikipedia, with around 100k question-answer pairs over 75,000 movie-related entities targeting factual reasoning over a curated movie KG. MetaQA builds upon WikiMovies by introducing paraphrased and more compositional questions. In the 1-hop setting, each question is grounded on a single KG fact, while the 2-hop version requires multi-hop inference to connect two supporting facts. Compared to WikiMovies, MetaQA includes more natural language variation and higher reasoning complexity, making it suitable for evaluating generalization under compositionality. Together, these datasets span both broad-domain and specialized-domain KGs, and cover reasoning from simple fact retrieval to multi-hop compositional queries.

### 4.2 BASELINES

We evaluate our method against several widely-used and representative baselines. **IO-prompt** (Brown et al., 2020) uses ChatGPT-3.5 to directly generate answers without the assistance of any external evidence, often resulting in hallucinations or factual errors due to limited LLM internal knowledge. **CoT-prompt** (Wei et al., 2022b) improves upon this by encouraging the model to generate intermediate reasoning steps, but its performance is still bounded by the LLM's pretraining knowledge. **Self-Consistency** (Wang et al., 2022) further extends CoT by sampling multiple reasoning paths and selecting the most frequent answer, improving robustness but still lacking access to external facts. **Sim-Retrieve** (Baek et al., 2023) retrieves KG triplets by computing embedding similarity between questions and candidate triplets using LLaMA2-7B or Flan-T5-3B. While simple and efficient, it often retrieves noisy or irrelevant facts, especially under multi-hop queries. **GNN-Scoring** introduces a graph neural network (GNN) to score and filter triplets with a lightweight language model encoding the query within the input of the GNN, but it still struggles with long-range dependencies and lacks semantic adaptability. **Think-on-Graph (ToG)** (Sun et al., 2024) improves retrieval reasoning by iteratively selecting and scoring subgraphs using LLaMA2-7B, Flan-T5-3B, or ChatGPT-3.5, and generating answers with ChatGPT-3.5. While effective, it incurs a higher computational cost and still depends heavily on the size of LLMs. In contrast, our method uses lightweight models to guide scoring and retrieval, achieving accurate reasoning without relying on expensive models.

Table 1: Experimental results (accuracy in %) with standard deviation of MoRA and baselines on QA datasets. We group methods into: retrieval-free prompting, ChatGPT retrieval, and lightweight LM retrieval using LLaMA2-7B or Flan-T5-3B. The best within each lightweight LM group are shown in **bold**, respectively. The best result of all the compared baselines is shown in underlined.

| Category | Method | WebQSP | WikiMovies | MetaQA-1hop | MetaQA-2hop |
|---|---|---|---|---|---|
| Retrieval-free | IO-prompt | $62.32 \pm 0.41$ | $69.24 \pm 0.35$ | $60.02 \pm 0.60$ | $23.01 \pm 0.79$ |
| | CoT-prompt | $62.74 \pm 0.46$ | $72.45 \pm 0.42$ | $64.98 \pm 0.52$ | $26.71 \pm 0.88$ |
| | Self-Consistency | $61.11 \pm 0.53$ | $73.64 \pm 0.37$ | $71.38 \pm 0.39$ | $30.52 \pm 1.01$ |
| ChatGPT retrieval | ToG | $\underline{74.24} \pm 0.91$ | $\underline{95.77} \pm 0.63$ | $\underline{96.33} \pm 0.77$ | $\underline{94.43} \pm 0.89$ |
| LLaMA2 retrieval | Sim-Retrieve | $46.98 \pm 1.44$ | $50.09 \pm 1.11$ | $47.44 \pm 0.92$ | $21.02 \pm 1.67$ |
| | GNN-Scoring | $63.86 \pm 0.73$ | $78.32 \pm 0.48$ | $83.82 \pm 0.52$ | $71.63 \pm 0.91$ |
| | ToG | $66.70 \pm 0.85$ | $86.62 \pm 0.59$ | $88.63 \pm 0.67$ | $78.01 \pm 0.93$ |
| | **MoRA** | $\mathbf{75.02 \pm 0.88}$ | $\mathbf{98.29 \pm 0.76}$ | $\mathbf{97.25 \pm 0.99}$ | $\mathbf{94.91 \pm 0.72}$ |
| Flan-T5 retrieval | Sim-Retrieve | $44.78 \pm 1.35$ | $52.39 \pm 0.99$ | $47.93 \pm 0.85$ | $20.17 \pm 1.72$ |
| | GNN-Scoring | $64.56 \pm 0.65$ | $79.10 \pm 0.44$ | $84.69 \pm 0.49$ | $73.68 \pm 0.86$ |
| | ToG | $72.25 \pm 0.77$ | $94.80 \pm 0.78$ | $90.67 \pm 0.66$ | $89.60 \pm 0.98$ |
| | **MoRA** | $\mathbf{88.59 \pm 0.83}$ | $\mathbf{98.22 \pm 0.74}$ | $\mathbf{96.84 \pm 0.87}$ | $\mathbf{94.95 \pm 0.95}$ |

## 4.3 EXPERIMENTAL SETUP

To evaluate the effectiveness of our proposed framework, we conduct experiments using **ChatGPT-3.5** (OpenAI, 2022), **LLaMA2-7B** (Touvron et al., 2023b), and **Flan-T5-3B** (Wei et al., 2022a) as backbone language models for retrieval and answering. In our framework, LLaMA2-7B and Flan-T5-3B are used as retriever models for scoring and selecting KG triplets, while ChatGPT-3.5 is used only during the final answer generation stage for some baselines.

For triplet selection, the Mixture-of-Experts (MoE) module is configured with $K = 3$ experts by default. The multi-hop expansion process is controlled by a hop budget $H = 1$ for MetaQA-1hop and $H = 2$ for other datasets, balancing retrieval depth and noise. We cap the number of retrieved triplets to 20 per query to maintain prompt length constraints. Expert-specific and contextual soft prompts are jointly used to condition expert scoring and improve relevance under multi-hop contexts.

All experiments are conducted with 5 random runs, and we report the average accuracy across runs to mitigate variance. More implementation details can be found in Appendix B. We release our code at `https://anonymous.4open.science/r/MoRA`.

## 4.4 RESULTS AND ANALYSIS

Table 1 shows that MoRA consistently achieves the highest accuracy across all datasets and model backbones, significantly outperforming both retrieval-free prompting methods and alternative retrievers. Retrieval-free baselines such as IO-prompt and CoT-prompt fall short due to their lack of access to structured evidence. While embedding similarity-based retrievers often fail to model multi-hop dependencies, GNN-based retrievers cannot capture semantic alignment. Despite modeling relationships of question and knowledge, they provide limited flexibility in matching complex, compositional semantics. In contrast, MoRA demonstrates strong performance even under lightweight language models, outperforming ToG even when ToG uses ChatGPT as its retriever. This highlights the effectiveness of MoRA: the expert-specific soft prompts allow the model to capture diverse matching semantics, while the hop-aware contextual soft prompts maintain coherence across reasoning steps. Interestingly, Flan-T5-3B achieves substantially higher accuracy on WebQSP, suggesting that the encoder architecture like Flan-T5 is particularly well-suited for latent triplet scoring and multi-hop relevance estimation in the open domain. Importantly, MoRA requires no LLM finetuning or large-scale language model inference during retrieval, showing that compact models, when equipped with the right inductive biases, can also support accurate and robust multi-hop KG reasoning.

## 4.5 PARAMETER STUDY

We further study the impact of the number of experts $K$ in the Mixture-of-Experts module, focusing on WebQSP and WikiMovies with both LLaMA2-7B and Flan-T5-3B as the LLMs for retrieval. The results in Figure 3 show a clear upward trend when increasing $K$ from 1 to 3: the models consistently

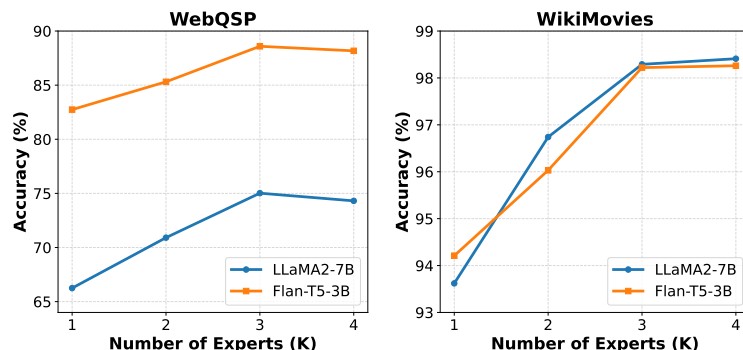

Figure 3: QA accuracy with varying number of experts $K$ in the MoRA retriever. We report results on WebQSP and WikiMovies using both LLaMA2-7B and Flan-T5-3B. Increasing $K$ from 1 to 3 improves performance across all settings, as more experts contribute diverse scoring perspectives. Performance saturates or slightly drops at $K = 4$, indicating that a small number of experts is sufficient for capturing the key semantic variations without unnecessary redundancy.

achieve better QA accuracy as more experts contribute complementary semantic views. However, the benefit quickly saturates beyond this point. In fact, while $K = 4$ performs slightly better than $K = 3$ on WikiMovies, the improvement is marginal, and on WebQSP with LLaMA2-7B the accuracy at $K = 4$ even drops compared to $K = 3$.

This observation suggests that having a moderate number of experts is sufficient to capture the main semantic diversity needed for effective scoring. Using too many experts introduces redundancy and increases training as well as inference cost, without yielding consistent performance gains. For this reason, we set $K = 3$ as the default configuration in our method, which achieves nearly the same accuracy as $K = 4$ while being more cost-effective and requiring lower computing cost.

### 4.6 ABLATION STUDY

We examine the impact of removing two key components: the expert-specific soft prompts and the contextual soft prompts. Results in Table 2 show that across all datasets and LLM settings, removing either consistently degrades performance. Without expert-specific prompts, the model lacks diversity for reasoning, leading to weaker discrimination especially on multi-hop datasets such as WebQSP and MetaQA-2hop where multiple relational patterns must be captured. Without contextual prompts, the model loses continuity across hops and struggles to select the right chain of evidence. This becomes severe in multi-hop reasoning QA settings. Notably, removing either type of prompt leads to consistent performance drops across both models. Removing expert-specific soft prompts results in a larger degradation on both models, particularly on complex reasoning settings like WebQSP, indicating its stronger reliance on specialized expert modeling. In contrast, the removal of contextual soft prompts hurts both models on multi-hop datasets, highlighting the importance of preserving inter-hop coherence for complex reasoning. Both components play complementary and indispensable roles in enabling robust multi-hop reasoning.

## 5 RELATED WORKS

### 5.1 RETRIEVAL-AUGMENTED GENERATION

Retrieval-Augmented Generation (RAG) (Lewis et al., 2020b) combines the parametric knowledge of pre-trained language models with external non-parametric memory to address knowledge staleness and hallucination issues. The framework retrieves relevant passages from external corpora and conditions generation on both the input query and retrieved context, achieving significant improvements on knowledge-intensive tasks. Early variants explored different conditioning strategies, with subsequent improvements including end-to-end training approaches (Izacard et al., 2023), dense passage retrieval optimization (Karpukhin et al., 2020), and fusion-in-decoder architectures (Izacard & Grave, 2021). Recent advances have addressed key limitations through sophisticated retrieval strategies: Self-RAG (Asai et al., 2024) enables adaptive retrieval decisions through reflection tokens, RAPTOR (Sarthi et al., 2024) constructs hierarchical tree structures via recursive clustering, Anthropic's Contextual Retrieval (Anthropic, 2024) reduces retrieval failures by up to 67% through

Table 2: Ablation results (accuracy in %) comparing the full MoRA model with variants that remove the expert-specific and/or contextual soft prompts, across two LLM models: LLaMA2-7B and Flan-T5-3B. We report accuracy on four datasets: WebQSP, WikiMovies, MetaQA-1hop (MQ-1hop), and MetaQA-2hop (MQ-2hop). The best results are shown in **bold**.

| Method | LLaMA2-7B | | | | Flan-T5-3B | | | |
|---|---|---|---|---|---|---|---|---|
| | WebQSP | WikiMovies | MQ-1hop | MQ-2hop | WebQSP | WikiMovies | MQ-1hop | MQ-2hop |
| Full Model | **75.02** | **98.29** | **97.25** | **94.91** | **88.59** | **98.22** | **96.84** | **94.95** |
| w/o Expert-Specific | 67.83 | 94.27 | 88.14 | 84.63 | 83.74 | 95.08 | 90.82 | 88.94 |
| w/o Contextual | 70.01 | 95.46 | N/A | 90.91 | 86.23 | 97.31 | N/A | 91.18 |
| w/o Both | 63.98 | 86.01 | N/A | 77.86 | 71.65 | 93.88 | N/A | 86.97 |

contextual chunk embedding, GraphRAG (Edge et al., 2024) leverages knowledge graphs for global reasoning, and CRAG (Yan et al., 2024) introduces retrieval quality assessment mechanisms. Other notable developments include HyDE for hypothetical document generation (Gao et al., 2023a), ColBERT-based late interaction models (Khattab & Zaharia, 2020; Jha et al., 2024), and agentic systems that embed autonomous agents for dynamic strategy management (Singh et al., 2025; Dong et al., 2025). However, most RAG frameworks are text-centric and do not fully leverage structured knowledge graphs, limiting their reasoning transparency and multi-hop capability.

### 5.2 MIXTURE OF EXPERTS

Mixture of Experts (MoE) has emerged as a powerful architecture for scaling model capacity while maintaining computational efficiency through conditional computation (Fedus et al., 2022; Cai et al., 2025). Originally proposed to divide problem spaces into homogeneous regions handled by specialized sub-networks (Jacobs et al., 1991), MoE has been successfully adapted to large language models where different experts specialize in distinct aspects of knowledge and reasoning. Recent advances in MoE architectures include Mixtral 8x7B (Jiang et al., 2024), which demonstrated that sparse MoE models can outperform dense counterparts on knowledge-intensive tasks, and Switch Transformer (Fedus et al., 2022), which scales to trillion-parameter models by routing tokens to specialized experts based on learned gating functions. However, MoE models excel at memorization-intensive tasks but struggle with complex reasoning compared to dense models (Jelassi et al., 2025), highlighting the need for more sophisticated expert specialization strategies. In knowledge graph reasoning specifically, MoMoK (Zhang et al., 2025) introduces relation-guided modality knowledge experts for multi-modal knowledge graph completion, demonstrating how expert specialization can be tailored to handle different relational contexts and modality-specific reasoning patterns. These works suggest that MoE can provide complementary strengths for KG-based reasoning, though effective specialization mechanisms remain an open challenge. Our work captures diverse semantic perspectives by experts, offering an effective way for KG-RAG.

### 6 CONCLUSION

In this work, we proposed a novel knowledge graph–augmented retrieval framework for LLM-based question answering called MoRA. Our approach introduces a Mixture-of-Experts scoring module that leverages both expert-specific and contextual soft prompts to evaluate candidate triplets, enabling robust multi-hop reasoning across knowledge graphs. Through the designed hop-aware retrieval and selection process, our method ensures contextual and adaptive evidence exploration while maintaining efficiency. Importantly, the design does not require fine-tuning large backbone language models; instead, it relies on a compact scoring architecture while still fully exploiting the capabilities of LLMs for final answer generation. Extensive experiments on the benchmark datasets demonstrate the effectiveness of our approach. It consistently outperforms strong retrieval-based baselines, including Think-on-Graph, even when the latter is powered by larger models such as ChatGPT. Our ablation studies further confirm the critical roles of expert-specific and contextual prompts in achieving state-of-the-art performance. Overall, our retrieval and scoring mechanisms can substantially enhance the reasoning ability of LLMs without relying on large backbone models. We believe that our MoRA framework provides a general and cost-effective paradigm for future research in knowledge-enhanced LLM reasoning.

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

## A    LIMITATIONS

Although MoRA achieves strong retrieval and QA performance using lightweight language models, there remain some limitations. The use of a Mixture-of-Experts (MoE) architecture introduces a computational cost that scales with the number of experts $K$. As $K$ increases, each candidate triplet must be scored multiple times by experts, resulting in higher inference latency and memory consumption. The cost-effectiveness of MoRA depends on careful tuning of $K$, especially for real-time or resource-constrained applications. Additionally, while our framework supports multi-hop reasoning through iterative expansion and contextual prompt propagation, its design assumes that useful evidence can be recovered through hop-wise traversal of the KG. In cases where the answer requires long or indirect relational paths, the fixed hop limit and local expansion strategy may miss relevant information. Finally, our current setup is tailored to well-structured, clean KGs; extending MoRA to operate robustly under incomplete, noisy, or large-scale heterogeneous graphs is a promising but non-trivial direction for future work.

## B    EXPERIMENTAL SETTING

### B.1    BACKBONE MODELS

Our retrieval module operates on top of lightweight pre-trained language models, enabling efficient and scalable scoring without relying on large generative models. We experiment with two representative models: **LLaMA2-7B** and **Flan-T5-3B**.

**LLaMA2-7B.**    LLaMA2 (Touvron et al., 2023b) is a family of decoder-only transformer models pre-trained on a mixture of publicly available corpora. We adopt the 7B chat variant without instruction tuning, as provided by the HuggingFace Transformers library. Despite its relatively small size, LLaMA2-7B offers strong general-purpose language understanding and is suitable for plug-in scoring tasks. In our setup, we use LLaMA2-7B as a feature extractor: only the soft prompts, connecting layers and lightweight expert routing modules are trained, while all LLM parameters remain frozen. As a decoder-only model, while encoding texts into embeddings, we use the latent output of the first token of LLaMA2.

**Flan-T5-3B.**    Flan-T5 (Wei et al., 2022a) is an instruction-tuned extension of the T5 encoder-decoder architecture. The Flan variant is trained on a large collection of prompting tasks, making it particularly effective under few-shot and instruction-based settings. We use the 3B variant (Flan-T5-3B) in encoder-only mode to encode the concatenated prompt and triplet input. Like LLaMA2, the LLM model is frozen throughout training, with only the soft prompts, connecting layers and the MoE scoring components updated.

**Soft Prompt Integration.**    For both models, we prepend soft prompts as continuous embedding vectors at the input layer. No modifications are made to the model architecture or tokenizer. During training, gradients are propagated through the soft prompts and the mixture-of-experts scorer, ensuring high efficiency and compatibility with existing pre-trained weights.

### B.2    TRAINING DETAILS

We train all models using the Adam optimizer with learning rate $1 \times 10^{-4}$, batch size 20, and linear learning rate decay over 5 epochs. Prompt parameters are initialized randomly and trained from scratch. Each expert uses two separate prompt embeddings: one expert-specific and one contextual. We train all experiments on 4 A100 GPUs (80GB) with mixed-precision training enabled via PyTorch. During inference, retrieval is performed up to $H = 1$ hop for the MetaQA1-hop dataset and $H = 2$ for others, with $M = 20$ triplets selected per hop. We set the number of experts $K = 3$. The dimension of the latent embeddings in our method is set to the same dimension as the retrieval LLM (2,048 for Flan-T5-3B and 4,096 for LLaMA2-7B). All KG triplets are preprocessed into a bidirectional index, which maps each entity to its outgoing and incoming relations. This index is loaded into memory as a dictionary for efficient candidate expansion at each hop.

### B.3 EVALUATION PROTOCOL

We follow standard entity-level accuracy for all datasets, considering a prediction correct if it matches any of the gold answers. Each model is evaluated with 5 different random seeds, and we report the average accuracy and standard deviation across runs. For fair comparison, all baseline methods use the same KG and entity linker. The knowledge graph used for WikiMovies and WebQSP is provided by (Sun et al., 2018), while for MetaQA we use the set from (Kim et al., 2023). ToG baselines are re-implemented with the same number of hops and selection budget as MoRA.

### B.4 IMPLEMENTATION INFRASTRUCTURE

Our implementation is based on PyTorch and Python 3.11.7, and all experiments are conducted on NVIDIA A100 GPUs with 80GB memory.

We release our code at `https://anonymous.4open.science/r/MoRA`.

## C DATASETS

We evaluate our method on widely-used knowledge-based QA benchmarks, covering both one-hop and multi-hop reasoning settings.

### C.1 WEBQSP

WebQSP (Yih et al., 2016) is a widely used benchmark for multi-hop question answering over knowledge graphs, featuring natural language questions that often require compositional and multi-step reasoning. The underlying knowledge base is Freebase (Bollacker et al., 2008), from which we extract a task-specific subgraph. This yields a manageable KG suitable for retrieval-based methods while preserving sufficient coverage for answering the queries. Since WebQSP provides only the final answer entities without annotated reasoning paths, it poses a latent multi-hop retrieval challenge. For all experiments, we identify seed entities in each question to find the gold paths from a query entity to an answer entity.

### C.2 WIKIMOVIES

WikiMovies (Miller et al., 2016) is a question answering dataset grounded in a movie-related knowledge graph, featuring factual questions about entities such as films, actors, directors, and genres. Some questions can be resolved via single-hop reasoning (e.g., "Who directed The Matrix?"), though some involve mild compositionality. We use the version of the dataset that provides gold topic entities during training and inference. This setting enables a focused evaluation of the scoring mechanism's ability to retrieve precise evidence in relatively simple KG contexts, serving as a diagnostic case for our retriever.

### C.3 METAQA-1HOP AND METAQA-2HOP

MetaQA (Zhang et al., 2018) is a large-scale synthetic QA dataset grounded in the same movie knowledge graph as WikiMovies. It introduces significant linguistic variability through paraphrased and template-generated questions. MetaQA-1hop consists of questions that require single-hop reasoning (e.g., "Who starred in Titanic?"), while MetaQA-2hop involves two-step reasoning chains (e.g., "Which movies feature actors who worked with Quentin Tarantino?"). Compared to WebQSP, the underlying KG is more controlled and noise-free, but the dataset poses unique challenges related to relation composition and generalization across templated question forms. We use the "vanilla" version of the dataset with gold entity annotations, applying exact string matching for answer supervision and evaluation.

## D PROMPT DESIGN

Our method incorporates both standard instruction-style prompts and soft prompt tuning to guide the retrieval process. Below we detail the design choices for each component.

## D.1 BASE PROMPT FORMAT

To evaluate the relevance of a candidate triplet with respect to a question, we adopt a minimal and general natural language template:

```
Please evaluate if the triplet is relevant to the
given question.
Question:  {question}
Triplet:  ({head}, {relation}, {tail})
```

This simple instruction provides consistent guidance to the language model across datasets without requiring task-specific templates. The prompt is used as the base for each expert in the scoring module.

## D.2 SOFT PROMPT INTEGRATION

We prepend two types of soft prompts to the input of each expert before passing it to the lightweight language model:

- **Expert-Specific Soft Prompt**: Each expert $k$ is assigned a unique learnable embedding of 5 tokens, denoted as $p_k^{\text{exp}}$, which captures its inductive bias and enables specialization across diverse reasoning types.

- **Contextual Soft Prompt**: At each hop $t$, every expert also receives a 1-token contextual prompt $p_k^{\text{cxt}}(t)$, dynamically updated based on its scoring output at the previous hop. This allows the model to retain and propagate reasoning context across steps.

These soft prompts are trained jointly with the expert scorers, and allow flexible control over retrieval behavior without modifying the base language model weights. In our implementation, soft prompts are treated as continuous embeddings and optimized via backpropagation.

## D.3 PROMPT-MODEL INTERFACE

All inputs including soft prompts, question text, and candidate triplets are concatenated into a single sequence and encoded by the underlying language model. We use either LLaMA2-7B or Flan-T5-3B as the backbone encoder, with soft prompt tuning applied via prepended embeddings. The model is trained using a cross-entropy objective over a binary relevance label for each triplet.

## D.4 ANSWER GENERATION PROMPT

For the final answer generation stage, we employ a simple instruction-style prompt to guide the LLM in producing entity-level answers from the retrieved evidence.

```
You are a knowledgeable assistant helping to answer
questions based on evidence.
Given the following context:
{evidence}
Answer the question below as accurately as possible.
If the answer is not in the context, make your best
guess.  Please return all the possible answers as a
list.  Given the reason of your thought.
Question:  {question}
Answer:
```

Here, {evidence} is replaced with the retrieved triplets and intermediate reasoning results, and {question} is the input query. This design ensures that the model grounds its output in retrieved KG evidence while still allowing limited generalization when the evidence is incomplete.

# E ALGORITHM

---

**Algorithm 1** MoRA Retrieval (per question $q$; inference-time scoring with learned prompts)

---

**Input:** KG $\mathcal{G}$, query $q$, hops $H$, top-$M$, experts $K$
**Input:** Frozen LM encoder; learned parameters $\Theta = \{ \{p_k^{\exp}\}_{k=1}^K, g, \{\mathbf{w}_k\}_{k=1}^K, W_r \}$
1: // *Notes:* $p_k^{\exp} \in \mathbb{R}^{L_{\exp} \times d}$ (e.g., $L_{\exp}$=5 tokens), $p_{\text{init}}^{\text{cxt}} \in \mathbb{R}^{L_{\text{cxt}} \times d}$ (e.g., $L_{\text{cxt}}$=1 token), $f$ is the encoding of LM encoder.
2: $\mathcal{E}^{(0)} \leftarrow \text{LinkEntities}(q), \quad \mathcal{V}^{(0)} \leftarrow \mathcal{E}^{(0)}, \quad \mathcal{T}_q^* \leftarrow \emptyset, \quad p_{\text{init}}^{\text{cxt}} \leftarrow \emptyset$
3: **for** $k = 1$ to $K$ **do**
4: $\quad p_k^{\text{cxt}}(1) \leftarrow p_{\text{init}}^{\text{cxt}}$
5: **end for**
6: **for** $t = 1$ to $H$ **do**
7: $\quad \mathcal{C}^{(t)} \leftarrow \text{ExpandCandidates}(\mathcal{G}, \mathcal{E}^{(t-1)})$
8: $\quad$ **for all** $x \in \mathcal{C}^{(t)}$ **do**
9: $\quad\quad$ **for** $k = 1$ to $K$ **do**
10: $\quad\quad\quad \mathbf{z}_k(q, x, t) \leftarrow \text{LM}([p_k^{\text{cxt}}(t); \, p_k^{\exp}; \, q; \, x])$
11: $\quad\quad\quad s_k(q, x, t) \leftarrow \sigma(\mathbf{w}_k^\top \mathbf{z}_k(q, x, t))$
12: $\quad\quad$ **end for**
13: $\quad\quad \alpha(q, x, t) \leftarrow \text{softmax}(W_r \, f(q, x))$
14: $\quad\quad S(q, x, t) \leftarrow \sum_{k=1}^K \alpha(q, x, t)[k] \cdot s_k(q, x, t)$
15: $\quad$ **end for**
16: $\quad \mathcal{T}_{\text{sel}}^{(t)} \leftarrow \text{Top-}M$ candidates in $\mathcal{C}^{(t)}$ by $S(q, \cdot, t)$
17: $\quad \mathcal{T}_q^* \leftarrow \mathcal{T}_q^* \cup \mathcal{T}_{\text{sel}}^{(t)}$
18: $\quad \mathcal{E}^{(t)} \leftarrow \{e' | (h, r, e') \in \mathcal{T}_{\text{sel}}^{(t)}\} \setminus \mathcal{V}^{(t-1)}$
19: $\quad \mathcal{V}^{(t)} \leftarrow \mathcal{V}^{(t-1)} \cup \mathcal{E}^{(t)}$
20: $\quad$ **if** $t < H$ **then**
21: $\quad\quad$ **for** $k = 1$ to $K$ **do**
22: $\quad\quad\quad \bar{\mathbf{z}}_k(t) \leftarrow \dfrac{1}{|\mathcal{C}^{(t)}|} \sum_{x \in \mathcal{C}^{(t)}} \mathbf{z}_k(q, x, t)$
23: $\quad\quad\quad p_k^{\text{cxt}}(t+1) \leftarrow g(\bar{\mathbf{z}}_k(t))$
24: $\quad\quad$ **end for**
25: $\quad$ **end if**
26: **end for**
**Output:** $\mathcal{T}_q^*$

---

# F REPRODUCIBILITY STATEMENT

We have taken several steps to make our work easy to reproduce. The full retrieval pipeline and scripts for experiments are released at our **anonymous repository**: `https://anonymous.4open.science/r/MoRA` (see Appendix B). The main paper specifies the task setup and model components (subsection 3.2), while **pseudocode** for candidate expansion, MoE scoring, contextual prompt updates, and the training loop is provided in the Appendix E. Exact **hyperparameters**, optimizer settings, expert counts, hop budgets, evidence budgets, and seed handling are reported in Appendix B (Backbone Models, Training Details, Evaluation Protocol). We document **dataset sources and preprocessing** for WebQSP, WikiMovies, and MetaQA in the Appendix C, and include the **prompts** used for retrieval scoring and answer generation in the Appendix D. To aid verification, we provide **ablation studies** and **parameter studies** (Tables/Figures in the main text). The repository contains environments, data loaders, and evaluation scripts.

# G THE USE OF LARGE LANGUAGE MODELS

In accordance with the ICLR policy on the use of large language models (LLMs), we explicitly state how LLMs were used in this work. Our research itself is centered on LLMs. We design methods

for LLM-based reasoning, run experiments with LLMs, and evaluate their behavior across multiple datasets. All scientific ideas, methodological contributions, theoretical analyses, and experimental designs in this paper were conceived, implemented, and validated entirely by the authors.

LLMs were employed solely as auxiliary tools for polishing the presentation of the paper. Specifically, they were used for minor improvements in grammar, phrasing, and readability during the manuscript preparation process. Importantly, LLMs were not used to generate novel ideas, design experiments, derive proofs, or conduct any analysis central to the research contributions. The intellectual content of this work remains the sole responsibility of the authors.

## H ETHICS STATEMENT

This research does not involve the collection or use of personal, sensitive, or identifiable data. All experiments are conducted on publicly available benchmark datasets (e.g., WebQSP, WikiMovies, MetaQA), which are commonly used in the QA and knowledge graph communities. The language models used in our framework are open-source and have been widely adopted in academic settings. We have taken care to ensure that our methodology does not propagate harmful biases or misinformation. Nonetheless, we acknowledge that any deployment of automated QA systems should consider potential risks such as factual inaccuracies and unintended misuse of the method.

