# OpenReview forum: "Mixture-of-Experts for Knowledge Graph Retrieval-Augmented Generation"
_ICLR.cc/2026/Conference — ICLR 2026 Conference Withdrawn Submission_

### Official Review · Reviewer_zfQf · 2025-10-22

**Soundness:** 2
**Presentation:** 2
**Contribution:** 2
**Rating:** 2
**Confidence:** 4

**Summary:**

This paper proposes Mixture-of-Experts for Knowledge Graph Retrieval-Augmented Generation (MoRA), a framework that leverages expert-specific and contextual soft prompts to enable lightweight language models to perform efficient, interpretable, and coherent multi-hop reasoning over knowledge graphs.

**Strengths:**

1. The paper’s writing is well-structured and logically coherent, clearly explaining the motivation, framework, and algorithm of MoRA.
2. The experiments are extensive and rigorous, including diverse datasets, strong baselines, and detailed ablation studies that convincingly demonstrate the advantages of both expert-specific and contextual soft prompts.

**Weaknesses:**

1. **Outdated model selection.** In Section 4.3 *Experiment Setup*, the backbone models used are ChatGPT-3.5, LLaMA2-7B, and Flan-T5-3B. These models are now considered obsolete within the research community. Using more advanced models such as GPT-4o or LLaMA 3.1 would provide higher reference value for related studies and facilitate broader adoption of this work.

2. **Severe lack of baseline comparisons.** The paper only compares MoRA with ToG among KBQA strategies integrated with LLMs. I strongly recommend that the authors include **RoG [1]**, **PoG [2]**, **G-Retriever [3]**, **GNN-RAG [4]**, and **ToG-2 [5]** as additional baselines for a more comprehensive evaluation against MoRA.

3. **Unexplained specialization of experts.** As mentioned in the Introduction, *“Each expert in the MoE … encouraging specialization in reasoning styles such as temporal reasoning or compositional logic.”* However, the mechanism of the Expert-Specific Soft Prompt is not adequately explained, leading to an inconsistency between the stated motivation and the presented implementation.

4. **Questionable capacity of the Contextual Soft Prompt to store reasoning context.** Only one token is used to encode reasoning state, yet observations from ToG suggest that contextual information grows rapidly during path search. Equation 8 employs a single MLP layer to project prior reasoning states into one token, which seems insufficient to ensure faithful preservation of the reasoning context.

[1]Luo L, Li Y F, Haffari G, et al. Reasoning on graphs: Faithful and interpretable large language model reasoning[J]. arXiv preprint arXiv:2310.01061, 2023.
[2] Chen L, Tong P, Jin Z, et al. Plan-on-graph: Self-correcting adaptive planning of large language model on knowledge graphs[J]. Advances in Neural Information Processing Systems, 2024, 37: 37665-37691.
[3] He X, Tian Y, Sun Y, et al. G-retriever: Retrieval-augmented generation for textual graph understanding and question answering[J]. Advances in Neural Information Processing Systems, 2024, 37: 132876-132907.
[4] Mavromatis C, Karypis G. Gnn-rag: Graph neural retrieval for large language model reasoning[J]. arXiv preprint arXiv:2405.20139, 2024.
[5] Ma S, Xu C, Jiang X, et al. Think-on-graph 2.0: Deep and faithful large language model reasoning with knowledge-guided retrieval augmented generation[J]. arXiv preprint arXiv:2407.10805, 2024.

**Questions:**

As discussed in Weakness.

---

### Official Review · Reviewer_aePV · 2025-10-22

**Soundness:** 2
**Presentation:** 4
**Contribution:** 2
**Rating:** 2
**Confidence:** 5

**Summary:**

This paper proposes MoRA, a mixture-of-expert structured relevant KG triples retrieval method for knowledge-based question answering. MoRA aims to address the limitation that LLMs are not good at assessing the relevance of candidate triples. In light of this, the manuscript introduces “Mixure-of-Expert scoring” module, which scores candidate KG triples based on the query and previously explored triples. The MoE scoring mechanism allows different experts to capture various key points of the question, ensuring stabler retrieval that may improve the question answering performance.

**Strengths:**

1.	This paper is clearly written and well organized.

2.	The manuscript accurately identifies the key limitation of previous KBQA methods, which is the retrieval recall of relevant triples.

3.	This reviewer believes the proposed MoE scoring approach can improve the triple scoring performance / stability of LLMs.

**Weaknesses:**

1.	The selected baseline methods are not up to date, which weakens the significance of the proposed method and the technical quality of the manuscript.

The proposed method is not compared with various baseline methods, such as RoG [1], GNN-RAG [2], SubgraphRAG [3], ToG2.0 [4], DoG [5], FastToG [6], GoG [7], PoG [8], KARPA [9] that are accepted to top-tier conferences.

Here, [1], [2], [3] are based on subgraph retrieval that are closely related to the proposed method MoRA. ToG2.0 [4] is a subsequent work of ToG that is mentioned and compared in the manuscript. [5-9] are recent non-concurrent works that have achieved nice performance on KBQA.

2.	The manuscript does not include the commonly adopted CWQ dataset for evaluation. The CWQ dataset contains 1-4 hop questions. Without the CWQ dataset (or equivalent one, like 2WikiMultiHopQA), one is not able to examine whether the proposed method can generalize to 3-4 hop scenarios.

3.	[Minor] LLaMA2 and Flan-T5 are somehow outdated. However, it is acceptable to use them if they can deliver adequate performance.

[1] Luo et al., Reasoning on Graphs: Faithful and Interpretable Large Language Model Reasoning (ICLR2024)

[2] Mavromatis and Karypis, GNN-RAG: Graph Neural Retrieval for Efficient Large Language Model Reasoning on Knowledge Graphs (ACL Findings 2025, available on Arxiv on May, 2024)

[3] Li, M. and Miao and Li, P., Simple Is Effective: The Roles of Graphs and Large Language Models in Knowledge-Graph-Based Retrieval-Augmented Generation (ICLR2025)

[4] Ma et al., Think-on-graph 2.0: Deep and faithful large language model reasoning with knowledge-guided retrieval augmented generation (ICLR2025)

[5] Li et al., Decoding on Graphs: Faithful and Sound Reasoning on Knowledge Graphs through Generation of Well-Formed Chains (ACL2025)

[6] Liang and Gu., Fast Think-on-Graph: Wider, Deeper and Faster Reasoning of Large Language Model on Knowledge Graph (AAAI2025)

[7] Xu et al., Generate-on-Graph: Treat LLM as both Agent and KG for Incomplete Knowledge Graph Question Answering (EMNLP2024)

[8] Chen et al., Plan-on-Graph: Self-Correcting Adaptive Planning of Large Language Model on Knowledge Graphs (NeurIPS 2024)

[9] Fang et al., KARPA: A Training-free Method of Adapting Knowledge Graph as References for Large Language Model’s Reasoning Path Aggregation (ACL Findings 2025)

**Questions:**

1.	This reviewer suggests the authors to move Table 2 to page 8.

2.	This reviewer is interested in seeing the performance of the proposed method on the CWQ dataset, which contains 1-4 hop questions, which is commonly adopted in evaluating various baseline methods.

3.	This reviewer suggests the authors to compare their proposed method with [1-9] mentioned in the weaknesses part.

---

### Official Review · Reviewer_S1QW · 2025-10-28

**Soundness:** 2
**Presentation:** 3
**Contribution:** 3
**Rating:** 6
**Confidence:** 3

**Summary:**

This paper propose a lightweight and efficient approach to retrieve relevant information for KG-RAG while maintaining contextual coherence across queries. To achieve this, the authors introduce MoRA, a Mixture-of-Experts-based retrieval framework that integrates adaptive and soft-prompting mechanisms without requiring fine-tuning of LMs.  Experimental results on several KG-RAG benchmarks demonstrate that MoRA achieves superior performance compared to classical approaches such as ToG.

**Strengths:**

- The use of soft prompting enhances retrieval to eliminating the need for LLM fine-tuning and controlling context length is interesting.

- Experimental results indicate that MoRA performs competitively or better than existing baselines, validating the effectiveness of the proposed approach.

**Weaknesses:**

1. Missing related work:  The related works section overlooks some relevant studies that also employ only lightweight LMs or directly modular LLM architectures for KG-RAG. For instance, Search-R1 [1] Subgraph-RAG[2].

2. Unclear necessity of the MoE design. While performance improves as the number of experts increases, it remains unclear whether the gains arise from the MoE architecture itself or merely from the increase in total model parameters. A more controlled comparison—such as comparing MoRA (3×Flan-T5-3B) with a single larger model (Flan-T5-11B)—would help isolate the contribution of the MoE mechanism.

[1] Search-R1: Training LLMs to Reason and Leverage Search Engines with Reinforcement Learning.
[2]Simple is effective: the roles of graphs and large language models in knowledge graph based retrieval augmented generation

**Questions:**

1. What are the computational costs of MoRA in terms of inference time, memory usage, and training cost compared to baseline methods?

2. Could the authors provide more analysis on the router of the MoE component? Such as, how does the router distribute different query types among experts, and some evidence that indicates the MoE structure is essential?

---

### Official Review · Reviewer_5jmG · 2025-10-31

**Soundness:** 3
**Presentation:** 3
**Contribution:** 3
**Rating:** 4
**Confidence:** 4

**Summary:**

This paper proposes MoRA, a lightweight framework that uses expert-specific and contextual soft prompts to enable efficient, accurate, and interpretable multi-hop knowledge retrieval for large language models.

**Strengths:**

This paper tackles the challenges of factual accuracy and multi-hop reasoning in knowledge graph–based retrieval by introducing the MoRA framework with expert-specific and contextual soft prompts. Extensive experiments and ablation analyses demonstrate that MoRA significantly outperforms existing methods, even those using larger language models.

**Weaknesses:**

1. Limited recall on long chains. A fixed hop budget and local expansion strategy can miss long KG paths, which constrains generalization performance on deeper multi-hop queries.

2. The paper claims that each expert specializes in distinct reasoning styles, yet it never explicitly maps or supervises any expert to a specific reasoning type, leading to conceptual inconsistency between design and implementation.

3. The comparison with the training-free ToG baseline is unfair, as MoRA involves additional training on soft prompts; a more thorough evaluation should include other trainable KBQA methods for a balanced comparison.

4. The paper introduces expert-specific and contextual soft prompts, but lacks comparison with training-free hard prompts, which are textual, fixed, and explainable. Thus, the advantage of soft prompts over hard prompts is not empirically verified.

**Questions:**

As discussed above in weakness.

---

### Note · Authors · 2025-11-26

I have read and agree with the venue's withdrawal policy on behalf of myself and my co-authors.